# Real-World Treatment Outcomes and Safety of Afatinib in Advanced Squamous Cell Lung Cancer Progressed after Platinum-Based Doublet Chemotherapy and Immunotherapy (SPACE Study)

**DOI:** 10.3390/cancers15235568

**Published:** 2023-11-24

**Authors:** Wonjun Ji, In-Jae Oh, Cheol-Kyu Park, Sung Yong Lee, Juwhan Choi, Jae Cheol Lee, Jiwon Kim, Seung Hyeun Lee

**Affiliations:** 1Department of Pulmonary and Critical Care Medicine, Asan Medical Center, University of Ulsan College of Medicine, Seoul 05505, Republic of Korea; jack1097@naver.com; 2Department of Internal Medicine, Chonnam National University Hwasun Hospital, Chonnam National University Medical School, Gwangju 58128, Republic of Korea; droij@chonnam.ac.kr (I.-J.O.); ckpark214@chonnam.ac.kr (C.-K.P.); 3Division of Pulmonary, Allergy, and Critical Care Medicine, Department of Internal Medicine, Korea University Guro Hospital, Korea University College of Medicine, Seoul 08308, Republic of Korea; syl0801@korea.ac.kr (S.Y.L.); choi133@korea.ac.kr (J.C.); 4Department of Oncology, Asan Medical Center, University of Ulsan College of Medicine, Seoul 05505, Republic of Korea; medljc@khu.ac.kr; 5Division of Pulmonary, Allergy, and Critical Care Medicine, Department of Internal Medicine, Kyung Hee University Hospital, Kyung Hee University College of Medicine, Seoul 02447, Republic of Korea; 14441@khmc.or.kr

**Keywords:** afatinib, squamous cell carcinoma, chemotherapy, immunotherapy, molecular profile, predictive biomarkers, real-world study

## Abstract

**Simple Summary:**

Although the LUX-Lung 8 trial has demonstrated the clinical benefit of afatinib as a second-line treatment for squamous cell carcinoma of the lung (LSCC), data on its use as a later-line treatment and sequential treatment following immunotherapy remain underexplored. This study on the real-world evidence of afatinib in LSCC patients who progressed both after chemotherapy and immunotherapy showed encouraging clinical outcomes with 2.1 months of time to treatment failure (TTF) and a 59.5% disease control rate in those patients without new safety signals. In addition, the erythroblastic oncogene B 2 (ERBB2) mutation was significantly associated with a longer TTF than the wild type. To the best of our knowledge, this is the first real-world study to demonstrate treatment outcomes, safety, and molecular biomarkers of afatinib in LSCC regardless of prior treatment lines. The present data may provide valuable insights for better management of patients with LSCC in routine clinical practice.

**Abstract:**

This study aimed to evaluate treatment outcomes and safety of afatinib in patients with squamous cell carcinoma of the lung (LSCC) who progressed after chemotherapy and immunotherapy. We recruited patients both retrospectively and prospectively and collected the outcomes and safety data. Additionally, we performed next-generation sequencing using tumor tissue and/or plasma to explore potential molecular biomarkers. Altogether, 42 patients were included in the final analysis. The median number of prior treatments was three (range 1–8), and the median TTF was 2.1 months. Objective response rate and disease control rate were 16.2% and 59.5%, respectively, and median duration of response was 4.0 months among response evaluable patients (*n* = 37). Treatment-related adverse events (TRAEs, including diarrhea, stomatitis, and paronychia) occurred in 22 (52.3%) patients; however, most were grade 2 or lower, and only 5 cases were grade 3. TRAEs led to dose modification in 17 (40.5%) and discontinuation in 4 (9.5%) patients. The TTF in patients with *ERBB2* mutations was significantly longer than that in patients without (6.8 vs. 2.1 months, *p* = 0.045). Our results highlight that afatinib is a reasonable treatment option in terms of effectiveness and safety, and *ERBB2* mutation can be used as a predictive biomarker in clinical settings.

## 1. Introduction

Although the relative incidence of squamous cell carcinoma of the lung (LSCC) to lung adenocarcinoma (LUAD) has decreased in many industrialized countries since 1990, LSCC is still a major histologic subtype constituting 20–30% of non-small cell lung cancer (NSCLC) [1,2]. Similar to other histological subtypes, approximately two-thirds of patients with LSCC are diagnosed at an advanced stage [3,4]. Despite recent breakthroughs in early diagnosis and therapeutics, there are limited treatment options compared to LUAD, and the prognosis is poorer [5].

Recently, immunotherapy, particularly programmed death-1 (PD-1)/programmed death ligand 1 (PD-L1) inhibitors, has revolutionized the treatment of lung cancer from early to advanced stages [6]. For LSCC, immunotherapy alone and a combinational approach with chemotherapy showed superiority over chemotherapy alone; however, the median overall survival (OS) in those patients treated with chemoimmunotherapy was only 17.2 months, and the 5-year OS rate was 18.4% [7,8,9]. Evidence has revealed that the genomic landscape of LSCC is unique compared to other subtypes of NSCLC. In contrast to LUAD, which harbors various driver genetic alterations, including the epidermal growth factor receptor (EGFR) and anaplastic lymphoma kinase in ~50% of the patients, LSCC is rarely associated with such drivers [10,11]. In addition, several unique candidate target pathways have been identified; however, the results of targeted therapy in LSCC remain disappointing [12], highlighting the unmet clinical need in this patient population.

Afatinib is an irreversible erythroblastic oncogene B (ERBB) family inhibitor that selectively blocks signaling from all homodimers and heterodimers formed by *EGFR*, *ERBB2*, *ERBB3*, and *ERBB4* [13]. In a pivotal phase III trial, second-line afatinib after platinum-based chemotherapy showed improved progression-free survival (PFS) and OS compared to erlotinib, along with a manageable safety profile [14]. The U.S. Food and Drug Administration (FDA) approved afatinib as a new oral treatment option for LSCC in April 2016. However, considering that immunotherapy-based regimens, either alone or in combination, are currently recommended as a frontline treatment, clinical data on afatinib use after the progression of such treatment and the efficacy and safety of later-line use of this agent are required to broaden its use in clinical practice.

To address this, we aimed to explore the real-world clinical evidence of afatinib in terms of clinical outcomes and safety in patients with LSCC in whom both chemotherapy and immunotherapy failed. In addition, we investigated the molecular profiles of patients to identify potential biomarkers. Our results would provide valuable evidence for the use of afatinib as sequential therapy and for response prediction and selection of more favorable patients before treatment, which may facilitate “personalized medicine” in this clinical scenario.

## 2. Materials and Methods

### 2.1. Study Design, Patients, and Data Collection

This is a retrospective and prospective, observational, multicenter study. Between April 2021 and March 2023, patients with advanced LSCC who progressed after (1) chemoimmunotherapy or (2) platinum-based chemotherapy followed by immunotherapy or (3) immunotherapy followed by platinum-based chemotherapy were recruited from four tertiary hospitals in the Republic of Korea. The inclusion criteria were as follows: (1) histologically confirmed advanced, metastatic, or recurrent LSCC; (2) history of frontline platinum-based doublet chemotherapy plus immunotherapy, frontline platinum-based chemotherapy followed by subsequent immunotherapy, or immunotherapy followed by platinum-based chemotherapy; (3) receiving afatinib as a second- or later-line treatment; and (4) age > 19 years. Patients who received prior EGFR-tyrosine kinase inhibitors (TKIs) were excluded.

For the retrospective cohort, we enrolled all patients who completed afatinib treatment between the date of afatinib approval by the Food and Drug Administration of Korea (28 August 2018) and the initiation of the study. Patients who were on afatinib treatment on the day of initiation of the study or those who started receiving afatinib after the initiation of the study were assigned to the prospective cohort. Written informed consent was obtained from all patients.

### 2.2. Treatment 

As per routine practice, patients received afatinib (40, 30, or 20 mg) once daily, as indicated by the approved drug label. Dose reduction was performed based on patient tolerance. Treatment was continued until the patient experienced a serious adverse event (AE), disease progression, death, or withdrew informed consent. Tumor response was assessed using computed tomography (CT) at minimum 8-week intervals after treatment and evaluated according to the Response Evaluation Criteria in Solid Tumors (RECIST) 1.1 [15]. Brain magnetic resonance imaging or positron emission tomography was performed when necessary.

### 2.3. Data Collection

Clinicopathological and survival data were collected from medical chart reviews. Molecular profiling using targeted next-generation sequencing (NGS) was performed using tumor tissues obtained at diagnosis and plasma samples (only from patients in the prospective cohort) obtained before the initiation of afatinib. Subgroup analyses were performed on various clinicopathological parameters and genetic alterations. Treatment-related adverse events (TRAEs) during afatinib treatment were collected and assessed based on the Common Terminology Criteria for Adverse Events (CTCAE Version 5.0, Bethesda, MD, USA). In addition, information on immune-related adverse events (irAEs) during previous immunotherapy was collected to evaluate the possible relationship between the re-emergence of irAEs and afatinib use.

### 2.4. Sample Collection and Preparation

Whole blood samples (8 mL) were collected in K2-ethylenediaminetetraacetic acid (EDTA) tubes (BD Diagnostics, New Jersey, NJ, USA) and subjected to centrifugation at 1000× *g* for 10 min to obtain plasma. To remove cellular debris, the sample was subsequently transferred to a 2 mL centrifuge tube and centrifuged at 16,000× *g* and 4 °C for 10 min. The supernatant was transferred to a fresh tube and stored in 1 mL aliquots in a refrigerator at −80 °C until use. Cell-free DNA (cfDNA) was extracted from isolated plasma samples using a QIAamp Circulating Nucleic Acid Kit (Qiagen, Hilden, Germany). Tumor DNA was extracted from sections (40 μm thick) of formalin-fixed, paraffin-embedded (FFPE) tumor tissue using the QIAamp DNA FFPE Tissue Kit (Qiagen, Hilden, Germany). The assessment for quality and quantity of tumor DNA and cfDNA was performed using a 4150 TapeStation System (Agilent, Santa Clara, CA, USA) and a Qubit 3.0 fluorometer (Thermo Fisher Scientific, Waltham, MA, USA), respectively.

### 2.5. Targeted NGS

Targeted NGS was performed using the Cancer-PRIME^TM^ (Clinomics Inc. Ulsan, Republic of Korea). This panel was designed to detect insertions/deletions and single nucleotide variants (SNVs) in 51 actionable genes. Candidate genes that were associated with FDA-approved therapies or reported clinical trials were included in this panel. The list of genes in this panel is presented in Appendix A. The performance of this NGS-based assay has been validated [16,17,18]. In brief, the quality and size of purified DNA from samples were assessed using a Bioanalyzer system (Agilent, Santa Clara, CA, USA). In addition, DNA concentration was analyzed using a dsDNA Broad-Range (BR) assay with a Qubit fluorometer (Thermo Fisher Scientific, Waltham, MA, USA). We used data from variants obtained in the DNA samples from healthy individuals (*n* = 30) and common SNVs detected in the whole-genome sequencing from healthy unrelated Korean individuals (*n* = 50) [19] in order to filter out potential sequencing backgrounds.

### 2.6. Variant Calling

For variant calling, we used the human genome sequence hg19 provided by the National Center for Biotechnology Information as a reference. Sequencing and data acquisition were performed using Torrent Suite software (5.8.0). Sequencing coverage analysis was performed using Coverage Analysis (5.8.0.1) plugins, and VCF files were generated using variantCaller (5.8.0.19) plugins. Finally, Ion Reporter (5.10.2.0) software was used for the annotations of the variants. We defined SNVs according to the following criteria: (1) number of total coverages ≥ 500; (2) Phred-scaled average evidence per read ≥ 10; and (3) variant allele frequency ≥ 1%. The false-positive variants were filtered out using the Korean Personal Genome Project database. In addition, we included only pathogenic or likely pathogenic variants in the final analysis after excluding variants of unknown significance or benign variants based on the COSMIC and OncoKB databases.

### 2.7. Outcomes and Statistical Analyses

The primary endpoint of this study was the time to treatment failure (TTF). Objective response rate (ORR), disease control rate (DCR), duration of response (DoR), progression-free survival (PFS), OS, and safety profiles were secondary endpoints. Differences in treatment outcome parameters with respect to genetic alterations were analyzed exploratively. The comparison of clinical characteristics of each group was analyzed using Fisher’s exact test or chi-square test, as appropriate. TTF was defined as the time from the start of afatinib treatment to treatment discontinuation for any reason, including treatment toxicity, disease progression, or death. ORR was defined as the percentage of patients who achieved a complete or partial response. DCR was defined as the percentage of patients who achieved a complete or partial response or stable disease. PFS and OS were defined as the interval from the first day of afatinib treatment until disease progression or death and the period from the first day of the treatment to death from any cause, respectively. Data from patients without tumor recurrence or death were censored at the last follow-up. Correlations between clinicopathological parameters and clinical outcomes were estimated by univariate analysis using the log-rank test, followed by Cox proportional hazards regression analysis. Parameters with *p* values < 0.1 in the univariate analysis were subsequently included in the multivariate analysis. The Kaplan–Meier method was used to estimate survival probability. A difference was defined as statistically significant if *p* < 0.05. Statistical analysis was carried out using SPSS version 21.0 (IBM Corp., Armonk, NY, USA).

## 3. Results

### 3.1. Clinicopathological Characteristics of Patients

During the study period, 59 patients underwent initial screening. Among them, 10 were excluded because they did not meet eligibility criteria. Seven more patients were excluded for the following reasons: hospital transfer (*n* = 4), loss to follow-up (*n* = 2), and withdrawal of consent (*n* = 1). The final analysis included 42 patients (18 retrospectively and 24 prospectively).

Table 1 summarizes the baseline characteristics of the study cohort at the initiation of afatinib treatment. The participants were all Koreans with a median age of 69 years (range, 41–85). A total of 42 patients (52.4%) were below 70 years of age, 38 (90.5%) were male, and 40 (95.3%) were current or former smokers. Regarding the Eastern Cooperative Oncology Group performance status (ECOG PS), 33 (78.6%) patients had a score of 0 or 1. A total of 41 (97.6%) were diagnosed with stage IV disease, 18 (42.9%) had metastases affecting three or more organs, and 8 (19.1%) and 8 (19.1%) patients displayed metastases in the brain and liver, respectively.

In terms of prior treatment history, the median number of preceding lines of systemic therapy was three (range 1–8). Notably, 13 (30.9%) had undergone four or more lines of treatment prior to afatinib. Concerning the types of systemic treatment, most patients (35/42, 83.3%) underwent platinum-doublet followed by immunotherapy (21 received atezolizumab, 11 received pembrolizumab, and 3 received durvalumab). Six (14.3%) received chemoimmunotherapy with pembrolizumab, paclitaxel, and carboplatin, and one (2.4%) received pembrolizumab followed by chemotherapy. PD-L1 expression was assessed using the 22C3 pharmDx assay, and the tumor proportion scores (TPS) were available for 41 patients, of whom 14 (33.3%) demonstrated high PD-L1 expression (TPS ≥ 50%). Regarding afatinib treatment, 32 (76.2%) patients initiated afatinib with the standard dose of 40 mg, whereas 10 (23.8) initiated with 30 mg. Eighteen patients (42.9%) required dose modifications based on their tolerability.

### 3.2. Treatment Outcomes

Table 2 summarizes the follow-up and efficacy data of the study population. After a median follow-up period of 8.6 months (range, 1.0–21.0) since the initiation of afatinib treatment, 41 patients had discontinued the drug while one patient was still on treatment. The reasons for discontinuation were disease progression (*n* = 27), AEs (*n* = 5), death during treatment (*n* = 4), and withdrawal of consent (*n* = 2). Of these patients, 17 (40.1%) received next-line treatment, and 19 (45.2%) received best supportive care without further treatment.

Among the population, response data were available for 37 patients, of whom 6 had a partial response, 16 had stable disease, and 15 had progressive disease; thus, ORR was 16.2% and DCR was 59.5%. For six patients who showed PR, the median DoR was 4.0 months (95% confidence interval (CI), 0.9–6.4). To evaluate possible factors affecting the clinical benefit of afatinib, we compared DCR among different subgroups of patients. DCR was consistent regardless of clinicopathologic parameters; however, patients with dose modification showed better DCR than those without, although it was nonsignificant (odds ratio (OR): 3.30, 95% CI: 0.79–13.64, *p* = 0.099, Appendix A).

Median TTF in overall population was 2.1 months (95% CI, 1.5–2.8). To identify the factors affecting TTF, we performed a TTF analysis based on various clinicopathological parameters. As shown in Appendix A, patients with dose modification showed significantly longer TTF compared to those without modification (3.0 vs. 1.5 months, hazards ratio (HR): 2.07, 95% CI: 1.08–3.97, *p* = 0.029). The clinical courses of all study subjects, from afatinib treatment to data cutoff, are summarized in Figure 1.

The median PFS of the study population was 2.6 months (95% CI: 1.8–3.8), and the median OS was 6.1 months (95% CI: 3.6–10.0). Appendix A shows the overall survival (OS) of the study population. OS remained consistent across various subgroups within the population, suggesting the absence of variables affecting OS in the univariate analysis. Notably, patients who started with a dose of 40 mg afatinib exhibited a trend toward longer OS compared to those who started with a dose of 30 mg (7.3 vs. 3.7 months, HR: 2.11, 95% CI: 0.93–4.77, *p* = 0.073).

### 3.3. Explorative Biomarker Analysis

To identify potential molecular biomarkers, baseline tissue and/or plasma samples were assessed for genetic alterations using NGS. The number of patients whose data were available for tissue only, plasma only, and both tissue and plasma were 28, 10, and 9, respectively. Data from 29 patients were included in the final analysis (Figure 2).

Individual genetic alterations in the biomarker-evaluable populations are shown in Figure 3. The most prevalent mutations were in tumor protein P53 (TP53, 55.2%), Notch receptor 1 (NOTCH1, 34.5%), and CDKN2A (20.7%). Mutations in the ERBB receptor family were found in 24.1% (7/29) of patients. Mutations in EGFR (ERBB1) were detected in four patients, mutations in ERBB2 were detected in four patients, and mutations in ERBB4 were detected in one patient.

We further investigated the genetic alterations associated with clinical outcomes in this biomarker cohort. As summarized in Table 3, only ERBB family mutations were associated with the outcome variables. All patients harboring ERBB mutations exhibited disease control and showed a trend of higher DCR than those with the ERBB wild type, although it was not statistically significant (100% vs. 60%, *p* = 0.068). In addition, ERBB mutations also resulted in a tendency toward an extended TTF compared to the wild type (5.9 vs. 2.4 months, *p* = 0.101). Notably, ERBB2 mutations were significantly associated with a prolonged TTF (6.8 vs. 2.1 months, *p* = 0.045). Table 4 summarizes the individual data of the 7 ERBB-positive patients.

### 3.4. Safety

The safety profile of our study population is summarized in Table 5. TRAEs of any grade were observed in 22 patients (52.4%). The most common TRAEs were diarrhea (16 patients (38.1%)), stomatitis (10 patients (23.8%)), paronychia (5 patients (11.9%)), acneiform dermatitis (4 patients (9.5%)), and pneumonia (2 patients (4.8%)). Single cases of elevated liver enzyme levels, fatigue, anorexia, and epigastric pain have been previously reported. Most TRAEs were grade 2 or less, and only five cases of grade 3 TRAEs occurred: two pneumonia, one stomatitis, and one paronychia. No grade 4 or 5 TRAEs were reported. TRAEs led to dose modification (dose interruption, reduction, or both) in 17 patients (40.5%) and to the discontinuation of therapy in 4 (9.5%).

Immediately prior to afatinib treatment, 18 patients received immunotherapy with or without chemotherapy, while 24 received chemotherapy. The frequency of TRAEs between the two groups was not significantly different (40.1% vs. 51.0%, *p* = 0.789). We also collected data on irAEs in patients who had prior immunotherapy use. The median time between the final immunotherapy dose and the initial administration of afatinib was 37.5 days (range, 11.5–328.6). Five irAEs were recorded during immunotherapy: peripheral neuropathy (*n* = 2), pneumonia (*n* = 1), hypothyroidism (*n* = 1), and Guillain–Barré syndrome (*n* = 1). All irAEs were grade 2 or lower in severity. Notably, no irAEs were observed during afatinib treatment.

### 3.5. Clinical Case

We present a representative case of response to afatinib treatment. A 68-year-old male, who had a smoking history of 40 pack-years, was diagnosed with LSCC with lung-to-lung metastasis in April 2021. Using primary tumor tissue, molecular studies of EGFR mutations, anaplastic lymphoma kinase fusion, and ROS proto-oncogene 1 fusion were performed using PNA-clamping-based polymerase chain reaction (PCR), fluorescent in situ hybridization, and reverse transcription PCR, respectively, before the initiation of frontline treatment. There were no driver genetic alterations, and PD-L1 expression was negative. The patient underwent sequential treatment with six cycles of gemcitabine plus carboplatin, four cycles of atezolizumab, and six cycles of paclitaxel. Upon progression, he received afatinib as a fourth-line treatment. A baseline chest CT scan before afatinib treatment showed a mass in the right upper lobe (RUL) of the lung, along with multiple nodules in the anterior and posterior segments of the left upper lobe (LUL) (Figure 4A). Afatinib treatment was initiated at a standard dose of 40 mg. The patient developed grade 2 acneiform dermatitis and grade 2 stomatitis after 3 weeks of treatment. Subsequently, the dose was adjusted to 30 mg, resulting in an improvement in those AEs to grade 1. The first CT scan performed 2 months after afatinib treatment showed reduced primary lesions and lung nodules (Figure 4B). A subsequent CT scan 2 months later indicated the disappearance of the LUL nodules, while the RUL mass remained stable (Figure 4C). A follow-up CT scan after 6 months of treatment showed a minimal increase in the RUL mass without a change in the LUL lesion (Figure 4D). Thus, the treatment was sustained for another 2 months until the main mass enlarged and new lymphangitic and pleural metastases emerged (Figure 4E). The patient’s best response was PR, and TTF and PFS both equated to 8.6 months. Next-generation sequencing using primary tumor tissue and plasma revealed an ERBB2 mutation (c.2454_2455insG), which explained the favorable clinical outcome of the patient.

## 4. Discussion

In this study, we observed the clinical outcomes of 2.1 months TTF and DCR of 59.5% with no new safety signals in patients with LSCC treated with afatinib. In addition, the ERBB2 mutation was significantly associated with a longer TTF than the wild type. To the best of our knowledge, this is the first study to demonstrate real-world treatment outcomes and safety of afatinib in LSCC patients who progressed after immunotherapy and chemotherapy regardless of treatment lines. Considering that the patients in this study were heavily treated, the clinical benefits and tolerability are meaningful, and the molecular profiling data indicated that mutations in the ERBB receptor family could be used as predictive biomarkers in this clinical context.

Afatinib is the first and only FDA-approved oral agent used for the treatment of LSCC. The approval was based on the LUX-Lung 8 trial, which demonstrated superior PFS (2.4 vs. 1.9 months, HR: 0.82, *p* = 0.041) and OS (7.9 vs. 6.8 months, HR: 0.81, *p* = 0.0077) of afatinib over erlotinib in stage IIIB/IV LSCC patients who progressed after platinum-doublet chemotherapy [14]. Although this trial demonstrated the efficacy and safety of afatinib in LSCC, further evidence is essential to gain deeper insight into its effectiveness, safety, and role in treatment decisions in real-world clinical scenarios.

To date, there is only one real-world dataset on afatinib in such a clinical setting. Kim et al. reported the characteristics and outcomes of patients with LSCC who received second-line afatinib after first-line chemoimmunotherapy in the United States [20]. In that study, the median time to treatment (TOT) was 7.3 months in the afatinib cohort, which was numerically higher than that in the chemotherapy cohort (4.2 months). Notably, this TOT was much higher than PFS in the historical trial (2.4 months) [14]. Although the results of this study provide a rationale for the use of afatinib in LSCC, it is important to acknowledge the distinctive characteristics of the study population. Most importantly, the study included a substantial proportion, approximately 35%, of patients with mixed histology. Although detailed data regarding mixed histology were not available, a significant proportion of adenosquamous carcinomas, a subtype associated with EGFR mutations and favorable clinical outcomes, could be included within the population. In fact, the frequency of EGFR mutations within the afatinib cohort in that study reached 39%, a notably higher percentage compared to that previously reported (<5%) in LSCC [21]. The LUX-Lung 8 trial had a mere 4% of mixed histology cases [14]. Consequently, a longer TOT could be attributed to the enrichment of a specific histological subtype, thereby challenging the broad application of these findings. Such observations emphasize the necessity for further investigations aimed at exploring more homogeneous populations.

To reflect routine clinical practice, the present study was conducted using a design that distinguished it from the pivotal trial and the aforementioned observational study in several aspects. First, we included patients who had undergone chemoimmunotherapy and those who had received chemotherapy and immunotherapy sequentially. Second, we enrolled patients who had been treated with afatinib as second-line therapy and beyond. Finally, to assess the utility of pure squamous histology, we excluded patients with mixed histology. Despite such potential challenges in the study design, the clinical outcome data in our study, including DCR, TTF, and PFS, are quite similar to those of LUX-Lung 8 (2.4 months of PFS and 51% DCR), while RR was numerically higher than that of the trial (6%). Moreover, the clinical benefits were consistent across various subgroups, including various types of prior treatments and different afatinib treatment lines. Notably, TTF was significantly longer in patients with dose modification than in those without modification in this study. Previous studies have reported equivalent or better responses and clinical outcomes in patients with LUAD with dose reduction during afatinib treatment [22,23]; however, no studies have explored such association in LSCC. Our findings were in line with the prior results on LUAD and prove that active dose modification during afatinib treatment can also be beneficial in LSCC. Notably, the OS data indicated a tendency toward longer OS in the standard-dose group, suggesting that the starting dose might impact long-term clinical outcomes. Taken together, beginning with a 40 mg standard dose while allowing for dose modification could be an optimal treatment strategy for LSCC treated with afatinib, although further large-scale studies are warranted to validate this hypothesis.

In addition to investigating clinical outcomes, we performed molecular profiling to identify potential molecular biomarkers for predicting clinical benefits. Previous studies have reported that LSCC is dependent on the ERBB pathway; ERBB1 (EGFR) is overexpressed in two-thirds of SCC cases, whereas ERBB2 and ERBB3 are overexpressed in one-third of the cases. In addition, gene copy-number alterations in EGFR were found in 7–10% of LSCC [24,25,26,27]. Although EGFR mutations are less common, 21.6% of SCC patients have at least one ERBB mutation [28,29,30]. These data suggest that ERBB-targeted therapy may be beneficial, at least in certain groups of patients with LSCC. Afatinib is a pan-ERBB inhibitor that irreversibly blocks ERBB family members by binding covalently. Preclinical data showed that afatinib exhibits antitumor activity in ERBB2-mutant lung cancer by downregulating the phosphorylation of ERBB2 and inducing G1 arrest and apoptosis [31]. In addition, increased ERBB2 phosphorylation and sensitivity to afatinib were observed in transfected Ba/F3 cells with ERBB2 mutations [32]. Moreover, since the first study reporting the clinical benefit of afatinib in ERBB2-mutant lung adenocarcinoma in 2011 [33], accumulating evidence has consistently demonstrated its clinical activity against various ERBB2-altered human malignancies, including lung, breast, and gastric cancers [34,35]. An ad hoc analysis of the LUX-Lung 8 trial, which investigated the association between clinical outcomes and ERBB family member gene alterations, demonstrated that PFS (4.9 vs. 3.0 months, HR: 0.62, *p* = 0.06) and OS (10.6 vs. 8.1 months, HR: 0.75, *p* = 0.21) were numerically better in patients with ERBB mutations than in those without mutations [28]. In our current investigation, we did not identify a significant correlation between molecular biomarkers and pure efficacy endpoints. Nevertheless, a significant correlation emerged between ERBB2 mutations and TTF. Specifically, TTF encompasses both treatment discontinuation due to toxicity and disease progression. Notably, our findings align with a previously mentioned historical study, which demonstrated that ERBB2 mutations were associated with a prolonged PFS following afatinib treatment compared to erlotinib. It is also pertinent to highlight that patients exhibiting ERBB family mutations exhibited a nonsignificant inclination toward improved DCR. This observation is consistent with earlier data indicating numerically more favorable clinical outcomes in patients with ERBB family mutations as opposed to those with the wild type [28]. While we cannot precisely elucidate why a significant association with molecular status was not observed in other efficacy endpoints, such as RR or OS, one potential contributing factor could be the limited sample size inherent in our study. Taken together, our data indicate that specific molecular characteristics can be used to predict treatment outcomes for afatinib in LSCC, highlighting the significance of employing NGS proactively, even in these populations.

The safety profile of afatinib is well documented and consistent with that of other EGFR-TKIs [36]. The key TRAEs of EGFR-TKIs include diarrhea, rash/acne, stomatitis, and paronychia, some of which appear to be more pronounced with afatinib [36]. However, these AEs were predictable and manageable with supportive care, treatment interruption, and/or dose modifications [37]. The observed TRAEs in our study were aligned with common AEs of EGFR-TKIs and did not reveal new or unexpected safety signals. In fact, a small proportion of patients (23.8%) started afatinib at 30 mg, while most of the patients started with the standard dose of 40 mg. TRAEs led to dose modification in 17 patients (40.5%) and to the discontinuation of therapy in 4 patients (9.5%). Compared to the historical trial, the dose adjustment rate was numerically higher, whereas the drug discontinuation rate was lower [14]. However, these rates were comparable with those of a recent real-world study involving Korean patients with LUAD treated with afatinib [38]. The lower drug discontinuation rate in this study can be partially attributed to initiating a lower starting dose in some patients and proactively adjusting the doses to manage toxicity. Our findings align with those of previous studies that demonstrated that a lower starting dose and active dose modification of afatinib reduced TRAEs without detracting from therapeutic efficacy in LUAD [39,40,41]. Although larger-scaled investigations are required, the strategy of commencing with a lower dose and actively modifying the doses appears to be effective for LSCC, enhancing patient outcomes while mitigating toxicity.

IrAEs during TKI treatment are a growing concern for sequential or concurrent regimens involving immunotherapy and TKIs [42,43]. A previous study highlighted that severe irAEs were common among patients who received PD-L1 inhibitors followed by osimertinib [44]. In that study, the occurrence of irAEs was influenced by the treatment sequence and the interval between immunotherapy and TKI treatment. Specifically, severe irAEs were notably absent in patients treated with osimertinib, followed by PD-L1 inhibitors, whereas most irAEs occurred in patients who received osimertinib within 40 days of cessation of immunotherapy. Notably, such irAEs have not been documented in patients treated with other EGFR-TKIs, such as afatinib and erlotinib, in that study [44]. This suggests that the risk of irAEs is not characteristic of the drug class but is specific to individual drugs. Nonetheless, the aforementioned real-world study on afatinib after chemoimmunotherapy in LSCC reported that severe irAEs (such as pneumonitis, hepatitis, and colitis) occurred in six patients during afatinib treatment, all of whom had previously experienced severe irAEs during previous chemoimmunotherapy [20]. In contrast, our study reported no irAEs during afatinib treatment. This disparity might be linked to the relatively mild irAEs during prior immunotherapy and the relatively longer interval between the two treatments within our cohort compared to the previous report. Collectively, these previous findings and our own suggest that afatinib can be safely used in patients without a history of severe irAEs. However, vigilant monitoring is crucial during afatinib administration, particularly in patients with previous severe irAEs.

This study has some limitations. First, the population was Asian, which restricts the generalization of our data on a global scale. Second, the sample size was small, and a considerable proportion of patients were enrolled retrospectively. Thus, selection bias or misclassification was inevitable. The enrolment challenges of this study were partially related to the fact that they coincided with the COVID-19 pandemic during the study period. In addition, afatinib for LSCC is not reimbursed in Korea, which can not only serve as a reason for clinicians hesitating to recommend afatinib to patients but also as a factor contributing to patient reluctance to use this drug. To mitigate these problems, we rigorously performed subgroup analyses and molecular profiling to identify potential factors that significantly affect survival outcomes. Using real-world data, the present study successfully validated the clinical benefits of afatinib in LSCC and potential molecular biomarkers suggested by previous studies. Despite these limitations, we believe that the insights gained from our data provide valuable contributions to the understanding of the clinical use of afatinib and could allow better patient management and improved survival outcomes in patients with LSCC.

## 5. Conclusions

We observed meaningful clinical benefits of afatinib with a manageable safety profile without new safety signals in our cohort, which is consistent with previous findings. Our results underscore the potential of afatinib as a valuable treatment option for patients with LSCC who experience treatment failure following platinum-based chemotherapy and immunotherapy. Additionally, our data emphasize the broader use of molecular profiling with NGS to identify patients who could experience enhanced benefits from afatinib in clinical practice. Further research is required to distinguish additional subgroups with favorable outcomes and formulate strategies, including combination approaches, to optimize the clinical advantages of this drug.

## Figures and Tables

**Figure 1 cancers-15-05568-f001:**
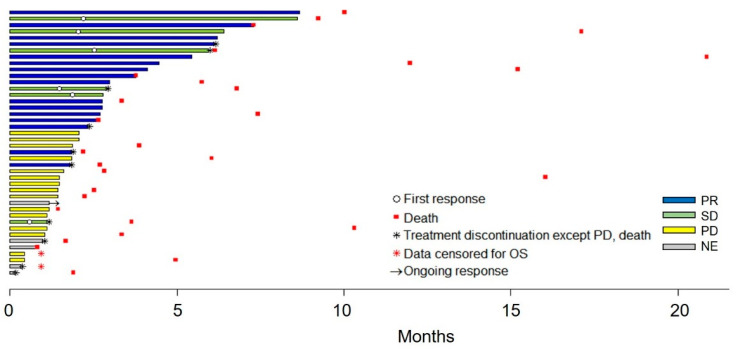
Clinical course of 42 study populations following afatinib treatment.

**Figure 2 cancers-15-05568-f002:**
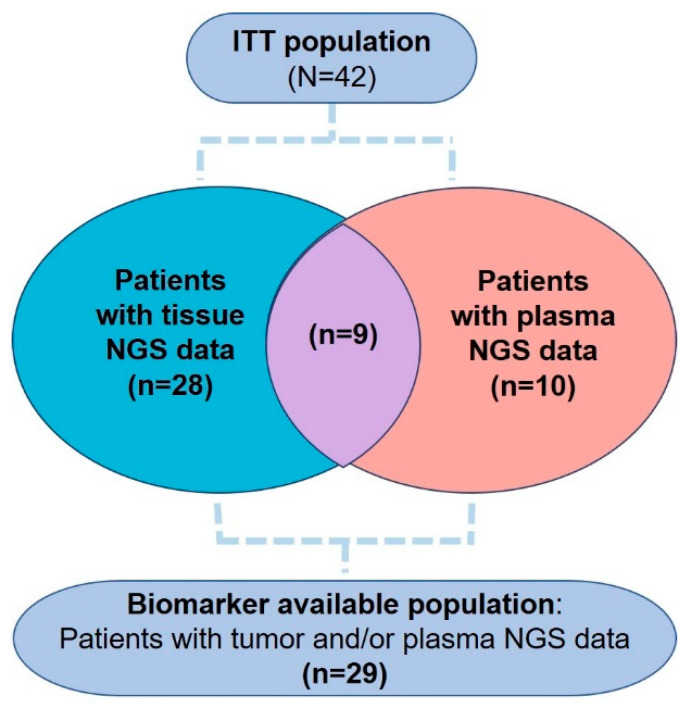
Patients enrolled for molecular biomarker analysis.

**Figure 3 cancers-15-05568-f003:**
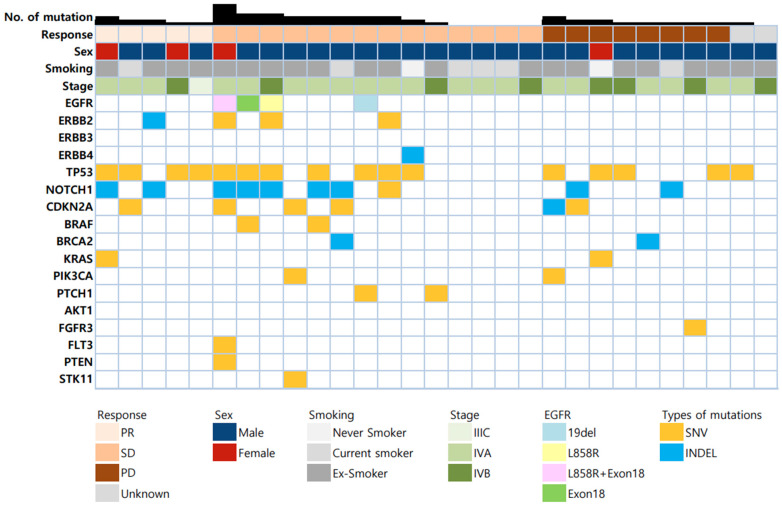
Genetic landscape of the biomarker-evaluable population (*n* = 29).

**Figure 4 cancers-15-05568-f004:**
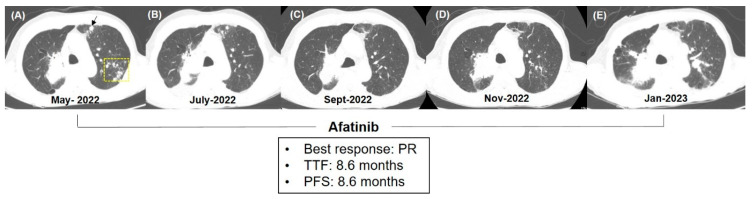
Clinical response to afatinib in a representative case. The patient was a 68-year-old male who was diagnosed with stage IVA with lung-to-lung metastasis. He sequentially received gemcitabine/carboplatin, atezolizumab, and paclitaxel before the fourth-line treatment of afatinib. Serial chest computed tomographic (CT) scan showed that the primary mass at the right upper lobe of the lung and nodules at the anterior (arrow) and posterior (yellow box) segments of the left upper lobe at baseline (**A**) gradually decreased on the response CT scans taken 2 (**B**) and 4 months (**C**) later. Follow-up CT scan taken at 6 months of treatment (**D**) showed only slightly increased primary mass, and thus, afatinib was maintained until the disease progression was apparent at 8 months (**E**). Best response for this patient was compatible with partial response, and both TTF and PFS were 8.6 months. PR, partial response; TTF, time to treatment failure; PFS, progression-free survival.

**Table 1 cancers-15-05568-t001:** Baseline characteristics of the patients (*n* = 42).

	No. of Patients (%)
Age (Years)	
Median (range)	69 (41–85)
<70	22 (52.4)
≥70	20 (47.6)
Sex	
Male	38 (90.5)
Female	4 (9.5)
Smoking status	
Never	2 (4.7)
Former	7 (16.7)
Current	33 (78.6)
Smoking intensity	
<30 pack-years	12 (28.6)
≥30 pack-years	30 (71.4)
ECOG PS	
0, 1	33 (78.6)
≥2	9 (21.4)
Differentiation	
Well, moderate	20 (47.6)
Poor	7 (16.7)
Unknown	15 (35.7)
Number of previous lines of systemic therapy before afatinib	
Median (range)	3 (1–8)
1	4 (9.5)
2	13 (30.9)
3	12 (28.6)
≥4	13 (30.9)
Types of previous systemic therapy	
Platinum-doublet followed by immunotherapy	35 (83.3)
Chemoimmunotherapy	6 (14.3)
Immunotherapy followed by platinum-doublet	1 (2.4)
Stage	
IIIC	1 (2.4)
IVA	29 (69.1)
IVB	12 (28.6)
Metastasis sites	
Brain	8 (19.1)
Lung	21 (50.0)
Liver	8 (19.1)
Bone	14 (33.3)
Pleura	17 (40.5)
Adrenal gland	1 (2.4)
Extrathoracic lymph nodes	22 (52.4)
Mediastinal lymph nodes	13 (30.9)
Number of involved organs	
<3	24 (57.1)
≥3	18 (42.9)
PD-L1 TPS	
<1%	13 (30.9)
1–49%	14 (33.3)
≥50%	14 (33.3)
Unknown	1 (2.4)
Afatinib starting dose	
30 mg	10 (23.8)
40 mg	32 (76.2)
Dose modification	
No	24 (57.1)
Yes	18 (42.9)

ECOG PS, Eastern Cooperative Oncology Group performance status; PD-L1, programmed death ligand 1; TPS, tumor proportion score.

**Table 2 cancers-15-05568-t002:** Efficacy of afatinib treatment (*n* = 42).

Variables	No. of Patients (%)
**Ongoing Treatment**	1
Reason for cessation of treatment	41
Progression	27
Adverse events	5
Withdrawal of consent	2
Death	4
Researchers’ decision	3
Best response	
CR	0 (0)
PR	6 (12.2)
SD	16 (32.7)
PD	15 (30.6)
NE	5 (24.5)
Objective response rate, %	16.2
Disease control rate, %	59.5
Follow-up duration, months, median (95% CI)	8.6 (1.0–21.0)
Duration of response, median (95% CI)	4.0 (0.9–6.4)
TTF, months, median (95% CI)	2.1 (1.5–2.8)
PFS, months, median (95% CI)	2.6 (1.8–3.8)
OS, months, median (95% CI)	6.1 (3.6–10.0)
Alive	10
Death	30
Lost to follow-up	1

Values are presented as the median (95% CI) or number. CR, complete response; PR, partial response; SD, stable disease; PD, progressive disease; NE, not evaluable; CI, confidence interval; TTF, time to treatment failure; PFS, progression-free survival; OS, overall survival.

**Table 3 cancers-15-05568-t003:** Genetic alterations affecting clinical outcomes in the biomarker-evaluable population (*n* = 29).

Mutations	No. of Patients	Disease Control		TTF		PFS		OS	
		DCR (%)	*p*-Value	Median (95% CI)	*p*-Value	Median (95% CI)	*p*-Value	Median (95% CI)	*p*-Value
EGFR			0.286		0.306		0.199		0.249
Negative	25	65.2		2.1 (1.6–3.0)		2.6 (1.9–3.8)		6.1 (2.8–10.0)	
Positive	4	100		5.2 (2.3-7.3)		5.9 (4.0-7.3)		15.2 (7.3–NE)	
ERBB2			0.286		0.045		0.134		0.781
Negative	25	65.2		2.1 (1.6–2.9)		2.6 (1.9–4.0)		6.8 (2.8–11.9)	
Positive	4	100		6.8 (3.8–8.6)		6.6 (3.8-8.6)		8.7 (3.8–NE)	
ERBB4			>0.999		0.868		0.483		0.167
Negative	28	69.2		2.5 (1.8–3.8)		3.5 (2.0–5.3)		6.8 (3.7–10.3)	
Positive	1	100		2.8 (NE–NE)		2.6 (NE–NE)		NR	
Any ERBB			0.068		0.101		0.113		0.156
Negative	22	60.0		2.0 (1.5–2.9)		2.4 (1.6–4.5)		6.0 (2.7–10.3)	
Positive	7	100		4.1 (2.3–7.3)		5.9 (2.6–8.6)		15.2 (3.8–NE)	

DCR, disease control rate; CI, confidence interval; TTF, time to treatment failure; PFS, progression-free survival; OS, overall survival; NE, not evaluable.

**Table 4 cancers-15-05568-t004:** Individual data of 7 patients with ERBB mutations.

Patient No.	ERBB Mutations	Sex/Age	Stage	Prev Lines of Tx	Starting Dose of Afatinib (mg)	ORR	DoR	TTF	PFS	OS	Reason of Discontinuation
S1P03	EGFR exon 18 and L858R ERBB2	F/69	IVA	8	40	SD	-	7.3	7.3	7.3	Death
S1P07	ERBB2	M/68	IVA	3	40	PR	6.4	8.6	8.6	9.2	PD
S1P12	EGFR exon 18	M/85	IVA	2	40	SD	-	2.3	4.9	4.9	Death
S2R14	ERBB4	M/73	IVA	2	40	SD	-	2.8	2.6	17.4	PD
S3P03	EGFR exon 19del	M/54	IVA	5	40	SD	-	4.1	4.0	15.2	PD
S3P06	ERBB2	M/75	IVA	5	40	SD	-	3.8	3.8	3.8	Death
S4P03	ERBB2	M/67	IVB	2	40	SD	-	6.2	5.9	16.9	PD

ORR, objective response rate; DoR, duration of response; TTF, time to treatment failure; PFS, progression-free survival; OS, overall survival; SD, stable disease; PR, partial response; PD, progressive disease.

**Table 5 cancers-15-05568-t005:** Treatment-related adverse events and tolerability during afatinib treatment.

	Any Grade	Grade 1 or 2	Grade 3	Grade 4	Grade 5
	No. of Patients (%)				
TRAEs *	22 (52.4)	19 (45.2)	3 (7.1)	0	0
TRAEs leading to dose modification	17 (40.5)	14 (33.3)	3 (7.1)	0	0
TRAEs leading to discontinuation of therapy	4 (9.5)	2 (4.8)	2 (4.8)	0	0
Types of TRAEs					
Diarrhea	16 (38.1)	16 (38.1)	0	0	0
Stomatitis	10 (23.8)	9 (21.4)	1 (2.4)	0	0
Paronychia	5 (11.9)	4 (9.5)	1 (2.4)	0	0
Acneiform dermatitis	4 (9.5)	4 (9.5)	0	0	0
Pneumonia/pneumonitis	2 (4.8)	0	2 (4.8)	0	0
Elevated liver enzyme	1 (2.4)	1 (2.4)	0	0	0
Anorexia	1 (2.4)	1 (2.4)	0	0	0
Itching	1 (2.4)	1 (2.4)	0	0	0
Nausea	1 (2.4)	1 (2.4)	0	0	0
Epigastric pain	1 (2.4)	1 (2.4)	0	0	0
Fatigue	1 (2.4)	0	1 (2.4)	0	0
irAEs	0	0	0	0	0

* The worst grade is reported for the patients with adverse events of multiple grades. TRAEs, treatment-related adverse events; irAEs: immune-related adverse events.

## Data Availability

The datasets used or analyzed in the current study are available from the corresponding author upon reasonable request.

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
