# Peer review of "Real-World Treatment Outcomes and Safety of Afatinib in Advanced Squamous Cell Lung Cancer Progressed after Platinum-Based Doublet Chemotherapy and Immunotherapy (SPACE Study)"

_cancers, 2023, doi:10.3390/cancers15235568_

Round 1

Reviewer 1 Report

Comments and Suggestions for Authors

The study "Real-world treatment outcomes and safety of afatinib in 2 advanced squamous cell lung cancer progressed after platinum-3 based doublet chemotherapy and immunotherapy (SPACE 4 study)" is a well documented research done to understand the efficacy of afatinib in terms of  treatment outcomes and safety squamous cell carcinoma of the lung. However, the patient cohort is small. Here are comments to improve it:

1. The cohort is too small to claim ERBB2 as a potential biomarker. Some in vitro studies or in vivo studies would be beneficial to add to support the claim. In vivo treatment with similar drug profile followed by validating ERBB2 status would help.  

Author Response

We thank you and the reviewers for your comments on our manuscript titled “Real-world treatment outcomes and safety of afatinib in advanced squamous cell lung cancer progressed after platinum-based doublet chemotherapy and immunotherapy (SPACE study)” (Manuscript ID: Cancers-2669722).

The manuscript has been revised in response to your thoughtful comments. We have included our point-by-point responses to the reviewers’ comments below. We hope that our manuscript is now suitable for publication in the Cancers as an Article.

Q1) The cohort is too small to claim ERBB2 as a potential biomarker. Some in vitro studies or in vivo studies would be beneficial to add to support the claim. In vivo treatment with similar drug profile followed by validating ERBB2 status would help.

  • Thank you for your valuable comment. It is well known that afatinib is a TKI, which has pan-ERBB activity and irreversibly blocks ERBB family members using the covalent bond. Preclinical data showed that afatinib exhibited antitumor activity in ERBB2-mutant lung cancer by downregulating the phosphorylation of ERBB2 and inducing G1 arrest and apoptosis. In addition, increased ERBB2 phosphorylation and sensitivity to afatinib were observed in transfected Ba/F3 cells with the ERBB2 mutation. Moreover, since the first study reporting the clinical benefit of afatinib in ERBB2-mutant lung adenocarcinoma in 2011, accumulating evidence has consistently supported the clinical efficacy of afatinib in various ERBB2-altered human malignancies, including lung, breast, and gastric cancers. An ad hoc analysis of the LUX-Lung 8 trial, which investigated the association between clinical outcomes and ERBB family member gene alterations, showed that ERBB2 mutations were associated with favorable PFS and OS following afatinib treatment. Taken together, preclinical and clinical data support the efficacy of afatinib in tumors harboring ERBB2 We added the information with appropriate references to the Discussion (lines 433-447).  

Reviewer 2 Report

Comments and Suggestions for Authors

This study aimed to evaluate treatment outcomes and safety of afatinib in patients with squamous cell carcinoma of the lung (LSCC) who progressed after chemotherapy and immunotherapy. The data is interesting  and it seems relevant to be published in cancers. Increased number of samples will be desirable but I recommend its publication in present form

Author Response

We thank you and the reviewers for your comments on our manuscript titled “Real-world treatment outcomes and safety of afatinib in advanced squamous cell lung cancer progressed after platinum-based doublet chemotherapy and immunotherapy (SPACE study)” (Manuscript ID: Cancers-2669722).

The manuscript has been revised in response to thoughtful comments of other reviewers. We hope that our manuscript is now suitable for publication in the Cancers as an Article.

Reviewer 3 Report

Comments and Suggestions for Authors

In their manuscript, Ji at al. present their results of real-world treatment outcomes of afatinib in advanced squamous cell lung cancer in a set of patients who progressed on previous chemotherapy and/or immunotherapy lines. The premise is based on the phase III LUX-Lung 8 trial, which compared afatinib to the first-generation EGFR inhibitor, erlotinib, with superior PFS and OS. However, the trial results were published in 2015. Since that trial was carried out, precision oncology has gained a much more pronounced prominence in clinical practice, and the presented approach goes contrary to the concept of molecularly-informed precision oncology. In patients with squamous cell NSCLC, the NCCN recommends considering molecular testing for EGFR and HER2 mutations, conducted as part of broad molecular profiling. Nevertheless, in this study, only 29 out of 42 patients had molecular testing at all. In line with the importance of molecularly-informed treatment considerations, this real-world study also concluded that HER2 mutant cases had significantly superior outcomes, but we can see similar trends with all other ERBB mutations (probably not significant due to low numbers of cases).

A major shortcoming of the manuscript, in terms of precision oncology, is a poor analysis of the molecular findings. There is no data considering the functional types of mutations, except EGFR mutations in Table 4, but even there, no specific mutations are provided. It would be critical to present whether the identified mutants are pathogenic or VUS (or benign), and to analyze the data in light of such mutation classification. In Figure 3, PR+CR and PD+SD are lumped together: they should be separated to uncover relations of outcomes with co-occurring mutations. The existence of possible resistance mechanisms, which could explain certain poor responses (e.g., MYC, ALK, PIK3CA, TSC1/2, BRAF, KRAS – to name just the most obvious ones) are not addressed in the analysis, and the reader is also unable to observe associations due to the lumped outcome info. Although the number of cases is rather limited, it would be worthwhile to perform subgroup analyses of cases with known resistance mechanisms vs. those without. Also, VUS alterations could be differentially indicated in Fig. 3 for clarity.

These concerns aside, the manuscript is generally well written and informative in terms of clinical aspects.

Specific remarks

Line 186: please give examples of exclusions due to researchers’ decision.

L. 199-200: the number of patients involved is 42, yet here a ratio of 35/41 is provided.

Figure 3: A surprisingly high number of patients had NOTCH1, FGFR1, ESR1, and PTCH1 mutations, although these are considered rare. This raises the possibility of either false positive callings or retaining common polymorphisms in bioinformatic filtering (which further underscores the importance of pathogenicity classification).

Case study: The durations of previous treatment lines in the case study should be presented. There is a discrepancy between the initial and later molecular testing, HER2 mutation was detected only by the latter (both in tissue and plasma then). What was the specific mutation? What could explain the discrepancy? (E.g., amplification of a HER2-mutated subclonal lineage during previous therapies.) What sample was used for the primary analysis?

Supplementary Table S2: Disease control rates are not presented as percentage (only numbers).

Supplementary Table S3 & 4: Case numbers are not presented.

Author Response

We thank you and the reviewers for your comments on our manuscript titled “Real-world treatment outcomes and safety of afatinib in advanced squamous cell lung cancer progressed after platinum-based doublet chemotherapy and immunotherapy (SPACE study)” (Manuscript ID: Cancers-2669722).

The manuscript has been revised in response to your thoughtful comments. We have included our point-by-point responses to the reviewers’ comments below. We hope that our manuscript is now suitable for publication in the Cancers as an Article.

Q1) In their manuscript, Ji at al. present their results of real-world treatment outcomes of afatinib in advanced squamous cell lung cancer in a set of patients who progressed on previous chemotherapy and/or immunotherapy lines. The premise is based on the phase III LUX-Lung 8 trial, which compared afatinib to the first-generation EGFR inhibitor, erlotinib, with superior PFS and OS. However, the trial results were published in 2015. Since that trial was carried out, precision oncology has gained a much more pronounced prominence in clinical practice, and the presented approach goes contrary to the concept of molecularly-informed precision oncology. In patients with squamous cell NSCLC, the NCCN recommends considering molecular testing for EGFR and HER2 mutations, conducted as part of broad molecular profiling. Nevertheless, in this study, only 29 out of 42 patients had molecular testing at all. In line with the importance of molecularly-informed treatment considerations, this real-world study also concluded that HER2 mutant cases had significantly superior outcomes, but we can see similar trends with all other ERBB mutations (probably not significant due to low numbers of cases).

  • Thank you for your valuable comment We agree with the reviewer’s suggestions. The current treatment landscape for lung cancer has changed dramatically since the adoption of immune checkpoint inhibitors, either alone or in combination, as frontline treatments for patients without driver genetic alterations. NCCN guideline currently recommends molecular testing even for squamous cell NSCLC. However, at the time of the conceptualization of this study (early 2019), molecular profiling was less widely adopted in squamous-type NSCLC than in adenocarcinoma, mainly because NGS was not reimbursed in our country. In addition, although IO-based regimens have been the standard frontline therapy for squamous cell NSCLC, little progress has been made in the treatment of squamous cell carcinoma over the last few decades. To address this clinical unmet need, we tried to investigate the clinical efficacy afatinib in the setting of real practice, and the results of Lux-Lung 8 trial can be applied to the current “IO-based frontline treatment era.” Although a previous real-world study evaluated the efficacy of afatinib in squamous cell NSCLC, the results cannot be directly translated to real-world practice for the reasons mentioned in the Discussion section. In this study, we aimed to investigate the usefulness of afatinib by recruiting patients who were treated with sequential chemotherapy after IO as well as frontline chemoimmunotherapy and patients treated with afatinib regardless of the treatment line, which reflects routine real-world practice. In addition, we attempted to identify potential molecular biomarkers. Although the sample size was small, we successfully identified a meaningful TTF of afatinib and a significant association between TTF and ERBB2 mutations, confirming the previous findings. Although we could not draw a firm conclusion from our data, we believe that afatinib is a reasonable option for the patient population, and a wider use of molecular profiling in squamous cell carcinoma is necessary to identify patients who could benefit from afatinib. Please consider these clinical implications of this study.

Q2) A major shortcoming of the manuscript, in terms of precision oncology, is a poor analysis of the molecular findings. There is no data considering the functional types of mutations, except EGFR mutations in Table 4, but even there, no specific mutations are provided. It would be critical to present whether the identified mutants are pathogenic or VUS (or benign), and to analyze the data in light of such mutation classification. In Figure 3, PR+CR and PD+SD are lumped together: they should be separated to uncover relations of outcomes with co-occurring mutations. The existence of possible resistance mechanisms, which could explain certain poor responses (e.g., MYC, ALK, PIK3CA, TSC1/2, BRAF, KRAS – to name just the most obvious ones) are not addressed in the analysis, and the reader is also unable to observe associations due to the lumped outcome info. Although the number of cases is rather limited, it would be worthwhile to perform subgroup analyses of cases with known resistance mechanisms vs. those without. Also, VUS alterations could be differentially indicated in Fig. 3 for clarity.

  • Thank you for your valuable comment We agree with the reviewer’s suggestions. We thoroughly reviewed molecular profile data and referred to the COSMIC and OncoKB databases to differentiate between pathogenic and VUS mutations. We included only pathogenic or likely pathogenic variants in the final analysis and have amended Figure 3 accordingly. In addition, we separated PR, SD, and PD using different colors. In Figure 3, we could see reduced frequencies of most mutations, but not ERBB family mutations. With these new data, we reevaluated whether other mutations were associated with the response to afatinib; however, we did not find any significant results. We have amended the Materials and Methods (lines 171–174) and Results (lines 275–276).
  •  

Q3) Line 186: please give examples of exclusions due to researchers’ decision.

  • Thank you for your valuable comments. We thoroughly reviewed the raw data and found that two patients were excluded because of hospital transfers and one was excluded because of loss to follow-up. We apologize for the error. Thus we have amended the corresponding sentence as follows: “Seven more patients were excluded for the following reasons: hospital transfers (n=4), loss to follow-up (n=2), and withdrawal of consent (n=1).” (lines 200–202).
  •  

Q4) L. 199-200: the number of patients involved is 42, yet here a ratio of 35/41 is provided.

  • It was a typographical error. We corrected 41 to 42. A percentage of 83.3% (35/42) did not need to be changed, and thus remained unchanged.

Q5) Figure 3: A surprisingly high number of patients had NOTCH1, FGFR1, ESR1, and PTCH1 mutations, although these are considered rare. This raises the possibility of either false positive callings or retaining common polymorphisms in bioinformatic filtering (which further underscores the importance of pathogenicity classification).

è Thank you for your valuable comments. As mentioned above, we included only pathogenic or likely pathogenic variants in the final analysis after excluding variants of unknown significance and benign variants based on the COSMIC and OncoKB databases. The false-positive variants were filtered using the Korean Personal Genome Project database. Finally, we have provided a new Figure 3 with a markedly reduced frequency of these mutations.

Q6) Case study: The durations of previous treatment lines in the case study should be presented. There is a discrepancy between the initial and later molecular testing, HER2 mutation was detected only by the latter (both in tissue and plasma then). What was the specific mutation? What could explain the discrepancy? (E.g., amplification of a HER2-mutated subclonal lineage during previous therapies.) What sample was used for the primary analysis?

  • Thank you for your valuable comment The number of cycles for the previous treatment has been added (lines 337–338). After pathological diagnosis, we only performed tests for EGFR mutations, ALK fusion, and ROS1 fusion using primary tumor tissue. After enrollment in this study, NGS was performed using the same tissue, and the tumor was found to harbor an ERBB2 mutation (c.2454_2455insG). We have added this information to the Case study (lines 332–336 & lines 352–353).

Q7) Supplementary Table S2: Disease control rates are not presented as percentage (only numbers).

  • Thank you for your valuable comment. We substituted these numbers into the disease control rates in Table S2.

Q8) Supplementary Table S3 & 4: Case numbers are not presented.

  • Thank you for your valuable comments. We have added the case numbers at the end of the title.

Reviewer 4 Report

Comments and Suggestions for Authors

There are few alternative treatments after chemo-immunotherapy in extensive pulmonary squamous cell carcinoma. Afatinib is the only targeted therapy that has demonstrated modest efficacy in this indication, but its impact on survival has never been evaluated in real life. This study therefore makes sense.

However, this study has many flaws, >which limit its scientific scop

>

>- its number is too small for a real-world study because this type of study requires a massive amount of data to be collecte

>

>- if the inclusion and exclusion criteria are well described, no details are given on how the patients were sought, particularly for the population included retrospectivly. In the absence of clear explanations on the recruitment method, we can question the existence of selection bia

>

>- a real-world study must prioritize effectiveness as the primary objective, which the TTF does not, because it includes tolerance in its definition. Overall survival should be preferred

>

>- molecular studies, although interesting in their concept, are not convincing, due again, to the small number of patients. For example, no conclusion regarding ERBB2 can be drawn (only 4 positive patients), especially that TTF is not a good en

>

>Note that the population being exclusively Asian, it does not reflect, on a global scale, the effectiveness of an anti-EGF

>

Author Response

We thank you and the reviewers for your comments on our manuscript titled “Real-world treatment outcomes and safety of afatinib in advanced squamous cell lung cancer progressed after platinum-based doublet chemotherapy and immunotherapy (SPACE study)” (Manuscript ID: Cancers-2669722).

The manuscript has been revised in response to your thoughtful comments. We have included our point-by-point responses to the reviewers’ comments below. We hope that our manuscript is now suitable for publication in the Cancers as an Article.

Q1) its number is too small for a real-world study because this type of study requires a massive amount of data to be collecte

A1) Thank you for your valuable comment. We agree with the reviewer’s comment. As mentioned at the end of the Discussion, the sample size was small. This is partly because of the overlap of our study period with the COVID-19 pandemic and the reimbursement issues in our country. During the study period (early 2019), afatinib was not reimbursed, and we cautiously expected reimbursement during the study period. However, afatinib for lung squamous cell carcinoma is not yet covered by the National Health Insurance Service (NHIS) of our country. Despite the small sample size, we performed rigorous analyses and molecular profiling to identify the potential factors associated with survival outcomes. Our data demonstrated a meaningful TTF for afatinib, even in our heavily treated population, and a significant association between TTF and ERBB2 mutations, both of which were in line with historical data (Lux-Lung 8 and its ad hoc study for genomic analysis). Although the sample size may not be sufficient to provide concrete evidence, we believe that our data highlight afatinib as a reasonable treatment option for squamous NSCLC subtypes, regardless of treatment line, especially in those harboring ERBB family mutations. Please consider the clinical implications of our study and unexpected situations encountered during the study period.

Q2) if the inclusion and exclusion criteria are well described, no details are given on how the patients were sought, particularly for the population included retrospectivly. In the absence of clear explanations on the recruitment method, we can question the existence of selection bia

A2) Thank you for your valuable comments. For the retrospective cohort, we enrolled all patients who completed afatinib treatment before the initiation of the study, regardless of the treatment line, since afatinib was approved by the Food and Drug Administration of Korea (August 2018). This has been clarified in the Materials and Methods section (lines 105–107).

Q3) a real-world study must prioritize effectiveness as the primary objective, which the TTF does not, because it includes tolerance in its definition. Overall survival should be preferred

A3) Thank you for your valuable comments. As you commented, overall survival could be an ideal primary endpoint for a real-world study. However, our patients were heavily treated (median 3 lines of prior treatment, range 1–8), thus, overall survival could be biased owing to the different time points of drug initiation in those patients. Instead, we selected TTF as the primary objective of this study, similar to many previous real-world studies. The median OS was 6.1 months and we have described the relevant data in the Results (lines 256–257) and Supplementary Table S4. We believe that our data on TTF and OS could provide valuable information for the management of patients with squamous cell carcinoma who would receive afatinib as a sequential treatment after the failure of immunotherapy, either alone or in combination.

Q4) molecular studies, although interesting in their concept, are not convincing, due again, to the small number of patients. For example, no conclusion regarding ERBB2 can be drawn (only 4 positive patients), especially that TTF is not a good en

A4) Thank you for your valuable comments. We agree with the reviewer’s comment. As mentioned at the end of the Discussion, the small sample size is partly due to the overlap of our study period and the COVID-19 pandemic and the reimbursement issue in our country. To compensate for this limitation, we rigorously performed subgroup analyses and molecular profiling to identify potential factors that significantly affect survival outcomes. Despite the small sample size, we demonstrated 2.1 months of TTF in our population and a significant association between TTF and ERBB2 mutations. Interestingly, these findings are consistent with historical data. Although our data may not be sufficient to provide concrete evidence for the real-world utility of afatinib in this clinical scenario, we believe our data suggest that afatinib is a reasonable treatment option for squamous NSCLC regardless of the treatment lines, especially in patients with ERBB2 mutations. Please consider the situations encountered during the study period and the clinical implications of our small but meaningful data.

Q5) Note that the population being exclusively Asian, it does not reflect, on a global scale, the effectiveness of an anti-EGF

A5) Thank you for your valuable comment We agree with the reviewer’s comment. Restriction to a specific race is another limitation of this study. To generalize our results, further studies enrolling patients of other ethnicities are required. We have added this point at the end of the Discussion (lines 493–494) as a limitation.

Round 2

Reviewer 1 Report

Comments and Suggestions for Authors

Thanks to the authors for responding. Although I still think it would be good to have further experiments to justify the claim and the cohort remains small, the added information in the discussion is improving the overall manuscript.

Author Response

The manuscript has been revised in response to your thoughtful comments. We have included our point-by-point responses to the reviewers’ comments below. We hope that our manuscript is now suitable for publication in the Cancers as an Article.

Q1) Thanks to the authors for responding. Although I still think it would be good to have further experiments to justify the claim and the cohort remains small, the added information in the discussion is improving the overall manuscript.

A1) We sincerely appreciate the enhancement of our manuscript, thanks to your invaluable comments.

Reviewer 4 Report

Comments and Suggestions for Authors

Many of the arguments made by the author are valid. However, I think an essential correction needs to be made in the discussion. If TTF can be maintained as the primary objective, correlating molecular biology with this endpoint, which also includes toxicity, is not correct because only a pure efficacy endpoint (response rate, overall survival) must be used. I would be in favor of a correction on this subject, especially in the discussion  by simply talking about a statistical trend between molecular status and certain effectiveness criteria such as DCR.

Author Response

We thank your comments on our manuscript.  The manuscript has been revised in response to your thoughtful comments. We have included our point-by-point responses to the reviewers’ comments below. We hope that our manuscript is now suitable for publication in the Cancers as an Article.

Q1) Many of the arguments made by the author are valid. However, I think an essential correction needs to be made in the discussion. If TTF can be maintained as the primary objective, correlating molecular biology with this endpoint, which also includes toxicity, is not correct because only a pure efficacy endpoint (response rate, overall survival) must be used. I would be in favor of a correction on this subject, especially in the discussion by simply talking about a statistical trend between molecular status and certain effectiveness criteria such as DCR.

A1) Thank you for your valuable comments. We concur with your suggestions. In our present study, we did not observe a significant association between molecular biomarkers and pure efficacy endpoints. However, a significant association was found with TTF, which encompasses treatment discontinuation due to toxicity and disease progression. While we cannot precisely explain why the specific mutation was not associated with other efficacy endpoints but showed significance with TTF, the limited sample size in this study is one plausible reason. According to your valuable comments, we have underscored the statistical trend observed between molecular status and specific effectiveness criteria, such as DCR in patients with ERBB family mutations in the Discussion (lines 446-458). Once again, we sincerely appreciate that your valuable comments contribute significantly to the improvement of our manuscript.